# Transcriptomics and Metabolomics Analysis of *Sclerotium rolfsii* Fermented with Differential Carbon Sources

**DOI:** 10.3390/foods11223706

**Published:** 2022-11-18

**Authors:** Jia Song, Yu Qiu, Rui Zhao, Jiayi Hou, Linna Tu, Zhiqiang Nie, Jianxin Wang, Yu Zheng, Min Wang

**Affiliations:** 1State Key Laboratory of Food Nutrition and Safety, Tianjin Engineering Research Center of Microbial Metabolism and Fermentation Process Control, College of Biotechnology, Tianjin University of Science & Technology, Tianjin 300457, China; 2Key Laboratory of Chemical Biology and Molecular Engineering, Ministry of Education, Institute of Biotechnology, Shanxi University, Taiyuan 030006, China; 3Wisconsin Center for NanoBioSystems, School of Pharmacy, University of Wisconsin—Madison, Madison, WI 53706, USA; 4Pharmaceutical Sciences Division, School of Pharmacy, University of Wisconsin—Madison, Madison, WI 53706, USA

**Keywords:** *Sclerotium rolfsii*, exopolysaccharides biosynthesis, transcriptomics, metabolomics

## Abstract

**Highlights:**

**Abstract:**

Scleroglucan is obtained from *Sclerotium rolfsii* and is widely used in many fields. In this study, transcriptomics combined with metabolomics were used to study the global metabolites and gene changes. The results of the joint analysis showed that the DEGs (differentially expressed genes) and DEMs (differentially expressed metabolites) of SEPS_48 (fermented with sucrose as a carbon source for 48 h) and GEPS_48 (fermented with glucose as a carbon source for 48 h) comparison groups were mainly related to cell metabolism, focusing on carbohydrate metabolism, amino acid metabolism, and amino sugar and nucleoside sugar metabolism. We therefore hypothesized that the significant differences in these metabolic processes were responsible for the differences in properties. Moreover, the joint analysis provides a scientific theoretical basis for fungal polysaccharides biosynthesis and provides new insights into the effects of carbon sources on the production. As an excellent bioenergy and biological product, scleroglucan can be better applied in different fields, such as the food industry.

## 1. Introduction

*Sclerotium rolfsii* is classified as basidiomycetes, umbelliferae, and polyporus in fungi [1,2]. The scleroglucan [3,4] produced under given conditions consists of 3 β-D-(1→3)- glucosyl and 1 β-D-(1→6)-glucosyl branch constituting a repeating unit [5,6]. The EPS (extracellular polysaccharide) has many attractive properties, such as good stability, salt resistance, and high-temperature resistance. It is widely used in food processing, oil recovery, the medicine industry, and the cosmetics industry [7,8,9]. For example, due to its excellent thermal stability, it has been of interest in heated food manufacturing processes. At present, the research on *S. rolfsii* at home and abroad mainly focuses on reducing the cost of production and increasing the yield. Survase et al. [10] optimized the medium using a statistical method and added metabolic precursors, with the highest yield of *S. rolfsii* MTCC2156 being 22.32 g·L^−1^ [4]. Li et al. [11] reviewed the production of scleroglucan, the effects of the production, and the application in the petroleum industry.

In our previous work, we discussed the differences in yield, rheological properties, and structures of polysaccharides under differential carbon sources [12]. Combined with the previous results on the viscosity and yield of EPS, the EPS fermented with the most popular carbon sources, sucrose and glucose, was selected as the goal of the multi-omics research. In recent years, the formation mechanism, the effects of substrates on the yield and properties at the gene level, and the changes of metabolites are not very clear. Several bacterial polysaccharides have been reported and characterized, such as glucan [12,13], and fructan [14] homopolysaccharides, and hyaluronic acid [15,16] and xanthan gum [17] heteropolysaccharides. To summarize, the main steps of the anabolic pathways of different polysaccharides are as follows: sugar monomer absorption, glycosyl transfer, sugar nucleotide donor formation, synthesis, modification, polymerization, and the macromolecular transport of polysaccharides [18]. However, the research on the anabolism of fungal polysaccharides is not as detailed as that for bacterial polysaccharides. The research on the synthetic pathway is relatively less.

Transcriptomics can understand the status of genes expressed in a specific physiological state, as well as cell phenotypic information [19]. It was not enough to rely on one method for a complete view of microbial metabolism [20]. Therefore, we have added metabolomics to directly reflect the state of the organism. Wang et al. [21] mentioned that the multi-group analysis of the effect of metal salts on polysaccharides yield provides a more comprehensive understanding of the mechanism. Bai et al. [22] used transcriptomics and metabolomics in order to explore the mechanism of acids on *Staphylococcus aureus*. 

Therefore, using LC-MS combined with RNA-Seq, we studied the changes of global metabolites and genes in order to understand why differential carbon sources produced differences in properties when producing polysaccharides. Moreover, the joint analysis provides a scientific theoretical basis for fungal polysaccharides biosynthesis and offers new insights into the effects of carbon sources on the production. As an excellent bioenergy and biological product, scleroglucan can be better applied in different fields, such as the food industry.

## 2. Materials and Methods

### 2.1. Materials and Reagents

The following materials were used: yeast extract and tryptone (OXOID Ltd., Basingstoke, Hampshire, UK), sucrose, glucose, MgSO_4_·7H_2_O, NaNO_3_, K_2_HPO_4_, KCl, FeSO_4_, and citric acid (Tianjin Yaohua Chemical Plant, Tianjin, China). The reagents used were all analytical grade.

### 2.2. Fungal Strains and Culture Condition

The fungal strain *S. rolfsii* ATCC-15205 was preserved in the China Microbial Strain Preservation Center. The fungal strain was first inoculated into PDA liquid medium (100 mL/250 mL) and cultured at 28 °C, 220 rpm, for 5–7 days to obtain a seed liquid containing a large amount of mycelium. Then, the seed liquid was inoculated into the fermentation medium according to the inoculum amount of 5%. The pH was adjusted to 4–5, and cultivated at 28 °C, 220 rpm, for 48 h.

The formula of the fermentation medium was as follows [12]: the carbon source (sucrose, glucose) (50 g/L), yeast extract (1 g/L), NaNO_3_ (2.25 g/L), K_2_ HPO_4_ (2 g/L), MgSO_4_·7H_2_O (0.5 g/L), KCl (0.5 g/L), FeSO_4_ (0.05 g/L), and citric acid (0.7 g/L) were dissolved in 1 L of distilled water, and the fermentation medium was sterilized at 115 °C for 30 min.

### 2.3. Transcriptomics and Metabolomics Analysis

After 48 h of fermentation, the fermentation broth of *S. rolfsii* cultured with different carbon sources was centrifuged at 8000× *g* for 10 min at 4 °C to remove the supernatant and resuspend the bacteria with precooled PBS solution (8000× *g*, 2 min, 4 °C), repeated 3–5 times. A portion was quickly frozen in liquid nitrogen and stored at −80 °C for metabolomics analysis. In the other part, the mycelium was rapidly added to 800 μL~1 mL Trizol to prevent RNA degradation, frozen for 5~10 min in liquid nitrogen, and transferred to −80 °C for storage. Six tubes for each sample were collected.

#### 2.3.1. Transcriptomics

RNA-seq analysis was performed on six independent biological replicates of SEPS_48 (fermented with sucrose as carbon source for 48 h) and GEPS_48 (fermented with glucose as carbon source for 48 h), with GEPS_48 as a control group. The total RNA extracted from the samples. The concentration and purity of the extracted RNA were detected by a NanoDrop 2000. The integrity of the RNA was detected by agarose gel electrophoresis, and the RIN value was determined by an Agilent 2100. Using magnetic beads with Oligo (dT) for A-T base pairing with ployA, mRNA can be isolated from total RNA, while later adding a fragmentation buffer. Through magnetic bead screening, mRNA can be broken down randomly. Small fragments of about 300 bp were isolated. Using the obtained mRNA fragment as a template, the first strand of cDNA was generated by reverse transcription, and the second strand of cDNA was further synthesized by PCR. The double strand of cDNA was purified and eluted with EB buffer. Afterwards, the purified cDNA double strands were subjected to end-repair and sequencing adapters. The target fragments were collected and PCR amplified. After a PCR amplification of 15 cycles, the library was enriched and sequenced on the Illumina platform. The original sequence is filtered by TGICL software: redundancy was removed to obtain high-quality sequences. The high-quality sequences were assembled for the bioinformatic analysis of unigenes.

The gene expression abundance of the two samples was counted, and the discrete degree of the gene expression levels between the samples was observed. Based on the Unigene sequence assembled by Trinity and the ORF sequence predicted by TransDecoder, the assembly results were obtained by tools such as Blast, Diamond, and HMMER. The annotation information in the database was then integrated using Trinotate (Trinotate Release v3.0.2) to obtain comprehensive functional annotation results. Using the annotation results of Trinotate, the genes annotated in each of the GO terms were counted according to the Gene Ontology database (GO), and the differentially expressed genes (DEGs) in the comparison groups were compared by DESeq2 software. The genes with |log2Ratio| ≥ 1 and q < 0.05 were regarded as significant DEGs.

The DEGs were analyzed by the Kyoto Encyclopedia of Genes and Genomes (KEGG). Firstly, the DEGs were annotated, and then enriched into the KEGG pathways. Each pathway in KEGG was enriched and analyzed by a hypergeometric test, and the significant enrichment of the pathways in the DEGs were found according to q < 0.05 as the standard. The genes with |log2Ratio| ≥ 1.2 and q < 0.05 were regarded as DEGs.

#### 2.3.2. Real-Time PCR Analysis

In total, 0.1 g of mycelium was extracted and the total RNA was extracted using a total RNA extraction kit (Solarbio, Beijing, China). After the RNA was extracted according to the above method and verified to be qualified, cDNA was synthesized from the total RNA using a PrimeScript™ RT reagent Kit with gDNA Eraser kit (TaKaRa, Shiga, Japan).

The cDNA samples taken above were diluted 10 times by DdH_2_O as an RT-PCR template. β-tublin was used as an internal reference gene, and primer 5 software was used to design specific primer pairs for selected target genes for verification. The experiment was carried out with TB Green ^®^Premix Ex Taq GC (Perfect Real Time) kit. The thermal cycle curve was: step 1, denaturation at 95 °C/30 s; step 2, PCR (40 repeats) 95 °C/10 s, 60 °C/30 s; step 3, melting at 95 °C/15 s, 60 °C/1 min, 95 °C/15 s. The relative expression level of the mRNA of the target gene was expressed by 2^−∆∆^Ct value.

#### 2.3.3. Metabolomics

The samples were separated by LC-MS. The single component was ionized by the ion source of the high vacuum mass spectrometer, and the mass spectrogram was obtained according to the m/z. Finally, the qualitative and quantitative results of the samples were obtained by analyzing the mass spectrometry data of the samples. Preprocessing the data effectively reduced the influence of irrelevant factors in the data group on the experiment. 

According to the identification of mass spectrometry, all of the metabolites were compared with the KEGG database. The annotation information of the metabolites in the database was obtained. According to the expression of the metabolites among the different samples, multivariate statistical methods such as PCA analysis and PLS-DA analysis were used. The sample data after data conversion was simplified by ROPLS (RPackages) processing software. The VIP value of each metabolite was obtained to evaluate the similarity of the samples within groups and the differences between groups.

The differentially expressed metabolites (DEMs) were annotated and enriched to the KEGG database to obtain the classification information and pathways. In this study, the potential differential metabolites were screened according to VIP > 1.0, *p* < 0.05, and then the target DEMs were screened by the KEGG differential metabolites enrichment pathway.

## 3. Results

### 3.1. Transcriptomics Identification of DEGs between SEPS_48 and GEPS_48 Comparison Groups

In order to analyze the effect of the different carbon sources of *S. rolfsii* on the synthesis of exopolysaccharides, we analyzed the transcriptomics of the SEPS_48 and GEPS_48 comparison groups. Each sample was in triplicate.

The volcano map (Figure 1A) clearly reflected the expression of different genes between the two groups. The DEGs were screened by the DESeq software package. Based on |log2Ratio| ≥ 1 and q < 0.05. 3536, the DEGs were identified in SEPS_48 and GEPS_48 (1440 upregulated and 2096 downregulated, respectively). The heat map in Figure 1B showed an overview of the expression profiles of the DEGs. It could be seen that there were significant differences in the expression patterns between the SEPS_48 and GEPS_48 comparison groups. The Venn diagram (Figure 1C) showed that there were 12,113 DEGs co-expressed in the SEPS_48 and GEPS_48 comparison groups.

### 3.2. GO and KEGG Pathway Enrichment Analysis of DEGs between SEPS_48 and GEPS_48 Comparison Groups

The enrichment results for the SEPS_48 and GEPS_48 comparison groups are shown in Figure 2A. The DEGs were mainly enriched in molecular function (MF). Among them, the terms with the most significant enrichment of DEGs were “Endopeptidase activity”, “Peptidase activity, acting on L-amino acid peptides”, and “Peptidase activity”. In the biological process (BP), the terms mainly related to “Proteasome regulatory particle assembly”, “Proteasome-mediated ubiquitin-dependent protein catabolic process”, and “Proteasomal protein catabolic process”. In the cellular component (CC), the most abundant term was “Proteasome regulatory particle, base subcomplex” and most of the genes in it were downregulated.

To further explore the functions of DEGs, we performed KEGG annotation and enrichment analysis. It can be seen from Figure 2B that most of the DEGs were annotated in the first category of metabolism, especially about “Carbohydrate metabolism”, “Amino acid metabolism”, “Energy metabolism”, and “Translation”, and “Folding, sorting and degradation folding, classification and degradation” in Genetic Information Processing. Cellular Processes were involved in “Transport and catabolism transport and catabolism”. Figure 2C showed that the DEGs were involved in “N-Glycan biosynthesis”, Lysine biosynthesis”, “Various types of N-glycan biosynthesis”, “Fatty acid biosynthesis”, “Proteasome”, “Peroxisome”, “Histidine metabolism”, “Alanine, aspartate and glutamate metabolism”, and so on. The metabolic pathways that enriched a large number of DEGs were “Protein processing in endoplasmic reticulum”, and the metabolic pathways with a high enrichment degree were “Alpha-Linolenic acid metabolism”, “Linoleic acid metabolism”, and “Lysine biosynthesis”.

### 3.3. Metabolomics Identification of DEMs between SEPS_48 and GEPS_48 Comparison Groups

In order to analyze the effect of differential carbon sources of *S. rolfsii* on the synthesis of EPS, we analyzed the metabolomics of the SEPS_48 and GEPS_48 comparison groups. Each sample was analyzed in triplicate. To further intuitively distinguish the grouping trend between the different samples and to understand the inter-group and intra-group differences, we carried out principal component analysis (PCA), as shown in Figure 3A,B. The score was 35.80% in positive ion mode and 24.70% in negative ion mode. The repeatability between the sample groups was good. The sampling points of the data were all within the 95% confidence interval, indicating that the data obtained were highly reliable. The PLS-DA plots in Figure 3C,D show that significant biochemical changes occurred between the samples. This suggested that VIP analysis can be used to screen for DEMs.

Therefore, under the condition that VIP >1 and *p* < 0.05 calculated by *t* test, 152 DEMs were detected in Figure 4A, of which 59 were upregulated and 93 downregulated. Consistent with the PCA, the heat map analysis of Figure 4B also showed significant differences between SEPS_48 and GEPS_48.

### 3.4. KEGG Pathway Enrichment Analyses of DEMs between SEPS_48 and GEPS_48 Comparison Groups

It can be seen from Figure 5A that most of the DEGs were annotated in the first category of metabolism, especially about “Amino acid metabolism”, “Nucleotide metabolism”, and “Lipid metabolism”. It was mainly about the process of “Membrane transport” in Environmental Information Processing. Figure 5B shows that DEMs were involved in “ABC transporters”, “Aminoacyl-tRNA biosynthesis”, “Alpha-Linolenic acid metabolism”, “Lysine biosynthesis”, and so on. The metabolic pathways that enriched a large number of DEMs were “Aminoacyl-tRNA biosynthesis”, “Purine metabolism”, “ABC transporters”, “Pyrimidine metabolism”, “Cyanoamino acid metabolism”, and the metabolic pathways with a high enrichment degree were “Aminoacyl-tRNA biosynthesis” and “D-Arginine and D-ornithine metabolism”.

### 3.5. Correlation Analysis of DEGs and DEMs between SEPS_48 and GEPS_48 Comparison Groups

To further investigate the differences in the biosynthetic pathways of EPS produced by different carbon sources, a correlation analysis of transcriptomics and metabolomics was performed.

Many genes and metabolites showed strong correlations between the SEPS_48 and GEPS_48 comparison groups (R > 0.8) (Appendix A). For example, there was a strong positive correlation between TRINITY_DN9515_c0_g4 (N-acetyl-gamma-glutamyl-phosphate reductase/acetylglutamate kinase), TRINITY_DN7956_c1_g2 (20 s proteasome subunit alpha 5), and uridine 5-diphosphate (C00015, enriched in pyrimidine metabolic) (R = 1 and *p* = 0 ≤ 0.001). There was a strong negative correlation between TRINITY_DN7685_c0_g2 (NADH-ubiquinone oxidoreductase chain), TRINITY_DN6538_c1_g1 (cell growth-regulating nucleolar protein), and L-arginine (C00062, enriched in arginine and proline metabolism and arginine biosynthesis) (R = 1 and *p* = 0 ≤ 0.001). It was suggested that these DEGs may play a direct or indirect role in regulating the metabolic pathway of related DEMs. The integrated analysis heat map in Figure 6B shows the association and expression between the DEGs and DEMs. A correlation network was established (Figure 6C), indicating the relationship between the DEGs and DEMs with strong correlations in the transcriptomics and metabolomics.

Among the SEPS_48 and GEPS_48 comparison groups, many DEGs and DEMs were co-annotated in the KEGG pathways, including pyrimidine metabolism (nucleotide metabolism) and purine metabolism (nucleotide metabolism), arginine and proline metabolism (amino acid metabolism), D-arginine and D-ornithine metabolism (amino acid metabolism), and amino sugar and nucleotide sugar metabolism (carbohydrate metabolism).

The most abundant KEGG pathway from the transcriptomics and metabolomics was the purine metabolic, which was related to eight DEMs and 78 DEGs (Appendix A). The second most abundant KEGG pathway was amino sugar and nucleotide sugar metabolism, which was related to eight DEMs and 63 DEGs (Appendix A). The third most abundant KEGG pathway was arginine and proline metabolism, which was related to five DEMs and 52 DEGs (Appendix A). Other enriched pathways were mostly related to membrane transport, lipid metabolism, translation, folding, classification, and degradation, such as ABC transporters, α-linolenic acid metabolism, and aminoacyl-tRNA biosynthesis.

### 3.6. qRT-PCR

In order to verify the transcriptome data, qRT-PCR was used to analyze the expression of the DEGs. With GEPS_48 as the control, more than 1 indicated that the gene expression was upregulated, otherwise it was downregulated. By calculating the relative expression of the verified DEGs, the expression levels of several DEGs were consistent with the transcriptome data (Figure 7). The results showed that the transcriptome data are reliable.

## 4. Discussion

*Sclerotium rolfsii* is the main fungus for the production of scleroglucan. Glucose and sucrose have always been the most popular carbon sources for the production of scleroglucan, both according to previous research [23] and our experimental results [12]. In recent years, the research on fungal polysaccharides was mainly focused on the optimization of fermentation conditions, separation and extraction, structure analysis, and activity evaluation. However, the research on the anabolism of fungal polysaccharides is not as detailed as that on bacterial polysaccharides, and the research on the synthetic pathway of polysaccharides is relatively less. At present, there is no research about the effects of different carbon sources on the genes, enzymes, and metabolites associated with exopolysaccharides synthesis in *S. rolfsii* using the correlation analysis of transcriptomics and metabolomics. Therefore, using LC-MS combined with RNA-Seq, we studied the changes of global metabolites and genes caused by EPS obtained from sucrose and glucose as carbon sources. In this study, a joint transcriptomics and metabolomics analysis was performed between the SEPS_48 and GEPS_48 comparison groups. It was found that nucleotide metabolism, amino acid metabolism, and carbohydrate metabolism played an important role in analyzing the potential relationship between DEGs and DEMs. We therefore hypothesized that the significant differences in these metabolic processes are responsible for the differences in properties. Moreover, they can not only provide a scientific theoretical basis for fungal polysaccharides biosynthesis, but also offer new insights into the effects of carbon sources on the production. As an excellent bioenergy and biological product, scleroglucan can be better applied in different fields.

Through the previous analysis, we analyzed the pyrimidine metabolism and purine metabolism of nucleotide metabolism. In the transcriptomics analysis of SEPS_48 and GEPS_48, we found that the *pold3, pole4, rpb5,* and *polr2e* genes, regulated as part of DNA polymerase subunits, were significantly upregulated, and most of the other genes were significantly downregulated, such as *codA, carB,* and *cpa2* (Appendix A). In the metabolomics analysis, uridine 5-monophosphate, 5′-CMP xanthine, uridine 5-diphosphate, and UDP-glucose were significantly upregulated, while cytosine, uridine, thymidine, adenosine, and xanthine nucleoside were significantly downregulated (Appendix A). Purine metabolism and pyrimidine metabolism were involved in processes such as the synthesis, degradation, recycling, interconversion, and transport of DNA, RNA, lipids, and carbohydrates. UMP (uridine 5′-diphosphate) exists as a precursor of pyrimidine nucleotides in pyrimidine metabolism, while nucleoside diphosphates and triphosphates are formed by nucleoside kinases [24]. The UMP in the pathway was significantly upregulated. UDP-glucose, as an important intermediate metabolite in previous hypothetical studies on the biosynthetic mechanism of scleroglucan [7], plays an important role in many different metabolic pathways including the biosynthesis of polysaccharides such as starch and glycogen, lipopolysaccharide, and glycosphingolipid. It was also significantly upregulated in this metabolic pathway. This indicated that the continuous accumulation of precursors was preparing for the synthesis of biomolecules, such as nucleic acids, lipids, and carbohydrates. Kusch et al. [25] also mentioned that participating in nucleotide synthesis or the protein cycle will increase in the exponential growth period in order to maintain the supply of nucleotide. It was shown that in the stage of rapid growth, nucleotide metabolism was active. The yield of the SEPS group was higher than that of GEPS; thus, it could be inferred that the metabolic activity of nucleotides was more vigorous. Moreover, most of the DEGs about DNA polymerase subunits regulated by *pole4, rpb5,* and *polr2e* were also significantly upregulated, indicating that purine metabolism and pyrimidine metabolism were involved in the synthesis and decomposition of DNA, RNA, and so on. The *ndk* and *nme* related to uridine 5-diphosphate-regulated nucleoside diphosphate kinase were significantly downregulated, while the associated metabolites were significantly upregulated. Huang et al. [24] showed that nucleoside diphosphates and triphosphates were formed by nucleoside kinases. Thus, we speculated that the pathways were not independent, but interacted with each other. The formation of uridine 5′-diphosphate was affected not only by one enzyme, but also by other enzymes. They were regulated by some genes. Some similar situations could also be inferred from this.

The amino acid metabolism was mainly about the arginine and proline metabolism, arginine biosynthesis, and the most significant enrichment was D-Arginine and D-ornithine metabolism. L-arginine was a differential metabolite in all three pathways. In the correlation network, it was also strongly correlated with DEGs from different pathways, indicating that L-arginine had a significant effect on the comparison groups. Interestingly, we found that among the three pathways, only *got1, gpt, alt,* and *arg56* were significantly upregulated in arginine biosynthesis, while the DEGs in other pathways were significantly downregulated. The amino acids in the differential metabolites of these pathways showed significant downregulation. When we analyzed the most relevant metabolite, L-arginine, we found that Hu et al. mentioned that N-acetyl-L-ornithine is the product of glutamic acid metabolism and can be used to synthesize ornithine in the study of the metabolic variation of *Rhizoctonia solani* [26]. Ornithine decarboxylates under the catalysis of decarboxylase to produce putrescine. When the growth of fungi stops due to glucose depletion, a sharp decrease in ornithine decarboxylase activity and putrescine content can be detected during the transition from mycelium to mature sclerotia [27]. Putrescine showed significant upregulation in the metabolic pathway. Therefore, we speculated that ornithine produced putrescine under the action of decarboxylase, which led to a decrease in the content of ornithine. Ornithine is produced by arginase in the pathway of arginine biosynthesis, but there was no significant difference in the genes regulated by this enzyme in the table of DEGs (Appendix A), and there was no significant difference in the genes and metabolites in the urea cycle in terms of arginine to ornithine. We speculated that the decrease in the arginine content was due to the decrease in ornithine. Arginine can not only be metabolized with amino acid proline and glutamic acid, but can also be used as the precursor of protein, nitric oxide, creatine, polyamine, agmatine, and urea [28]. It is important for the growth of cells, so we just inferred a certain conclusion of the analysis of the effect. The activity of various amino acid metabolic pathways during the combined analysis also indicated that cells were constantly using amino acids to synthesize and decompose proteins. The switching process, on the one hand, removes abnormal proteins, the accumulation of damaged cells. On the other hand, the activity of enzymes or proteins is regulated by synthesis and decomposition, thereby regulating cell metabolism.

The carbohydrate metabolism was mainly about the amino sugar and nucleotide sugar metabolism, galactose metabolism, and pentose and glucuronate interconversions. UDP-glucose was a differential metabolite in all three pathways. In the correlation network, it was also strongly correlated with DEGs from different pathways, indicating that UDP-glucose had a significant effect on the comparison groups and was an important intermediate in the synthesis of scleroglucan. Amino sugar and nucleotide sugar metabolism is a total pathway including the other two, achieved through the phosphorylation of D-fructose and a series of glucose isomerases and mutases. The enzyme reacted to obtain some important intermediates 1-glucose phosphate in this pathway, and then reacted with UTP to obtain UDP-glucose. In the SEPS_48 and GEPS_48 comparison groups, the genes involved in the process of UDP sugar synthesis were generally upregulated, such as *uap1, pgm3, chs1, manc, cpsb, glms, gfpt,* and *chs1*. This indicated that this pathway made an important contribution to the synthesis of scleroglucan. Amino sugars are mostly biological macromolecules, such as chitin, glycoprotein, lipopolysaccharide, or mucopolysaccharide [29]. For the synthesis of most polysaccharides, sugar nucleotides acted as glycosyl donors [30]. This study also detailed the synthesis and metabolism of nucleotide sugars. Nucleotide sugars are activated forms of carbohydrate synthesis or interconversion, such as DP-Gal or GDP-Fuc, a metabolite found in the pathway of aminosaccharide and nucleoside metabolism.

Finally, in the process of joint analysis, most of the metabolites in α-linolenic acid metabolism pathway of lipid metabolism were significantly upregulated, indicating that lipid metabolism was active. Gyamfi et al. [31] proposed that lipid metabolism was in a constant state of dynamic equilibrium. This means that some lipids were constantly oxidized to meet the cell’s metabolic needs, while others were synthesized and stored. The activity of this pathway suggested that the metabolic process produced fatty acids, which acted as part of the fungal cell membrane. The release of energy through oxidative decomposition can provide a large amount of ATP as energy for its growth and metabolism. The yield and viscosity of the SEPS group are both higher than GEPS group, so we speculated that it was also related to this pathway. 

In addition, biosynthetic pathways for amino sugars also involve the transfer of an amino group from an amino acid donor (usually L-glutamic acid or L-glutamine) to the keto functional group or carbon atom of a ketose derivative, and glutamate metabolism is the main network of ornithine, arginine, proline, and polyamine metabolic pathways. It is closely related to nitrogen metabolism. Pyrimidine metabolism is also partially involved in arginine biosynthesis. It is mainly related to the cell metabolism. In other words, amino acid metabolism, carbohydrate metabolism, and nucleotide metabolism influenced each other. Skarbek [29] et al. studied the GlcN-6-P synthase, which also appeared in the amino sugar and nucleoside sugar metabolic pathway. They mentioned a reaction that the enzyme can catalyze without any coenzyme, which involved the transfer of the amide group from L-glutamine to D-Fru-6-P. The ketose–aldose isomerization with the fructoimine intermediate indicated that the reactions of the three pathways interacted with each other.

## 5. Conclusions

In this study, we used LC-MS combined with RNA-seq to investigate the changes of global metabolites and genes during *Sclerotium rolfsii* cultured with sucrose and glucose as carbon sources. It was found that between the SEPS_48 and GEPS_48 comparison groups, the pathways that had significant effects were on carbohydrate metabolism, amino acid metabolism, and nucleotide metabolism. It is indicated that the EPS obtained by fermentation with sucrose and glucose had a significant effect on cell metabolism. We therefore hypothesized that the significant differences in these metabolic processes are responsible for the differences in properties. Moreover, the joint analysis provides a scientific theoretical basis for fungal polysaccharide biosynthesis and provides new insights into the effects of carbon sources on the production. As an excellent bioenergy and biological product, scleroglucan can be better applied in different fields, such as the food industry.

## Figures and Tables

**Figure 1 foods-11-03706-f001:**
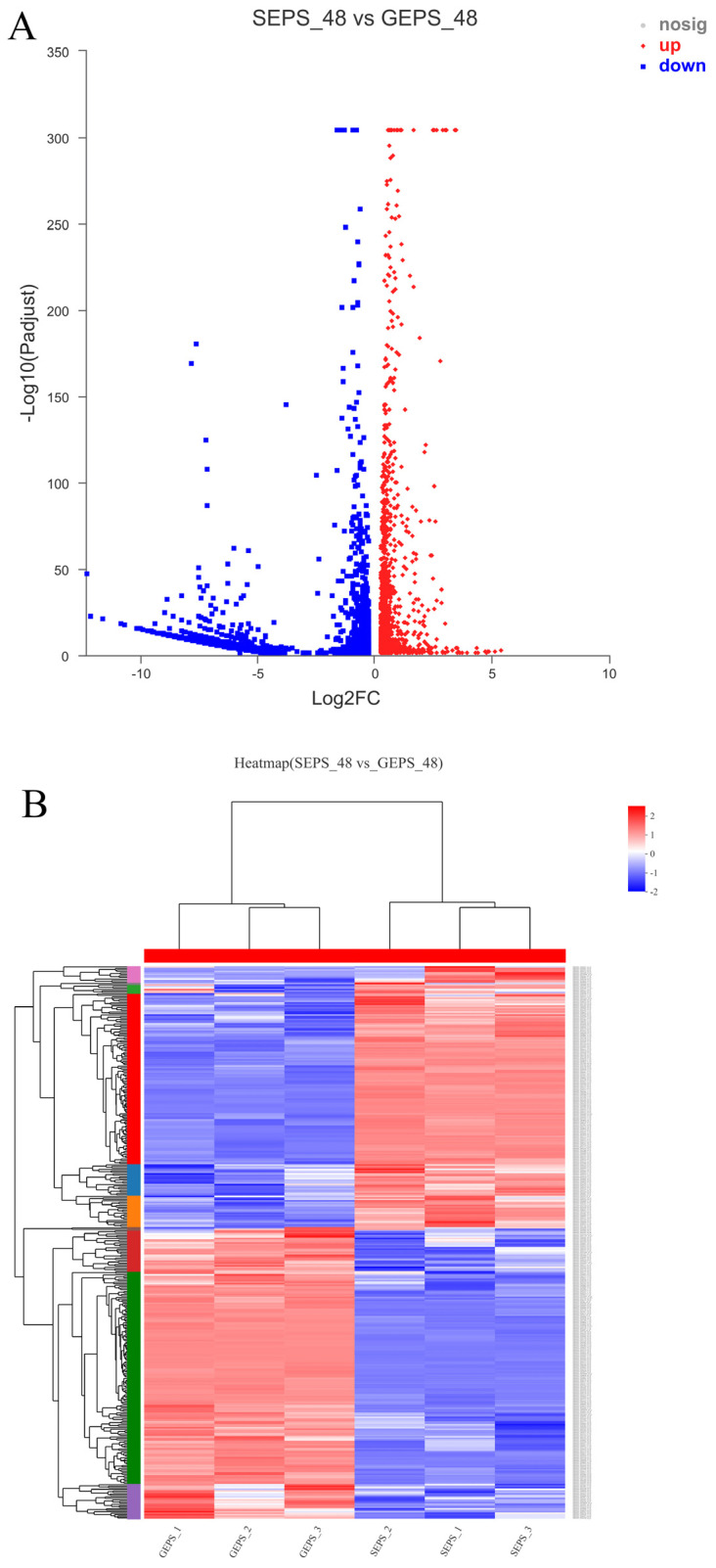
Analysis of DEGs in SEPS_48 and GEPS_48 comparison groups. (**A**) Number of up- and downregulated DEGs in the SEPS_48 and GEPS_48 comparison groups. Red dots represent upregulated DEGs, blue dots represent downregulated DEGs. (**B**) Heatmap of the DEGs in the SEPS_48 and GEPS_48 comparison groups. Red indicates highly expressed genes and blue indicates less-expressed genes. (**C**) Venn diagram of number of DEGs in SEPS_48 and GEPS_48 comparison groups.

**Figure 2 foods-11-03706-f002:**
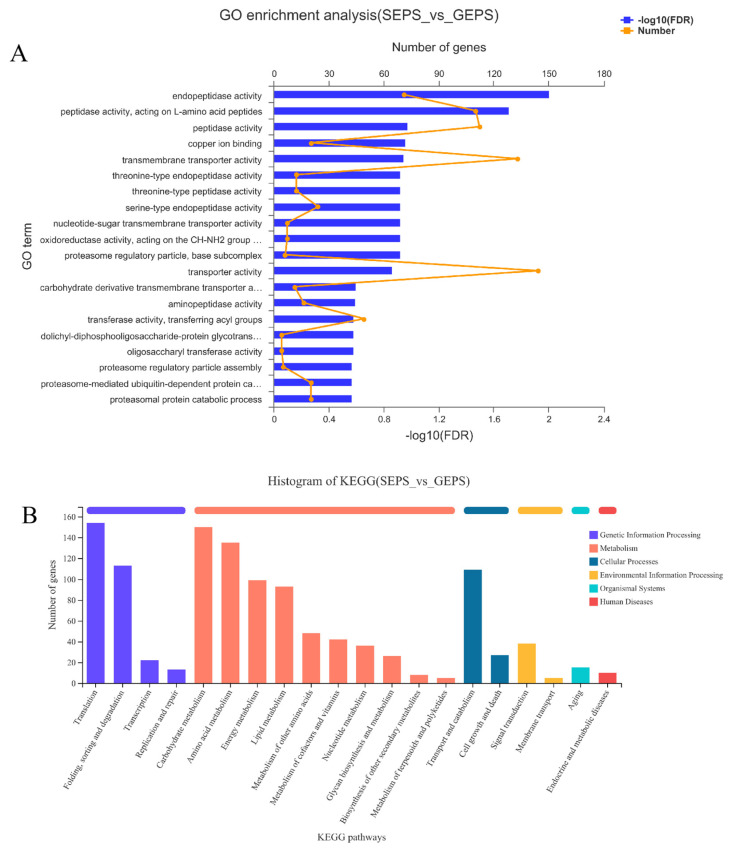
Enrichment analysis of DEGs between SEPS_48 and GEPS_48 comparison groups based on GO and KEGG pathway. The most enriched top 20 pathways terms. (**A**) GO enrichment column chart of DEGs. (**B**) KEGG annotation histogram of DEGs. (**C**) KEGG enrichment bubble chart of DEGs (the color represents the *p*-value, and the size of the bubble represents the number of enriched genes).

**Figure 3 foods-11-03706-f003:**
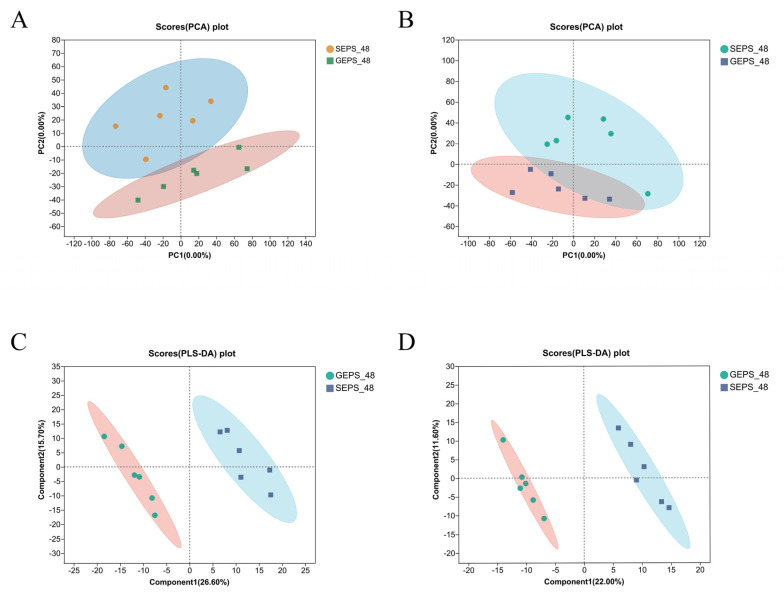
(**A**,**B**) PCA score chart of SEPS_48 and GEPS_48 comparison groups’ metabolite profile in positive and negative ion modes. (**C**,**D**) PLS−DA score chart of SEPS_48 and GEPS_48 comparison groups’ metabolite profile in positive and negative ion modes.

**Figure 4 foods-11-03706-f004:**
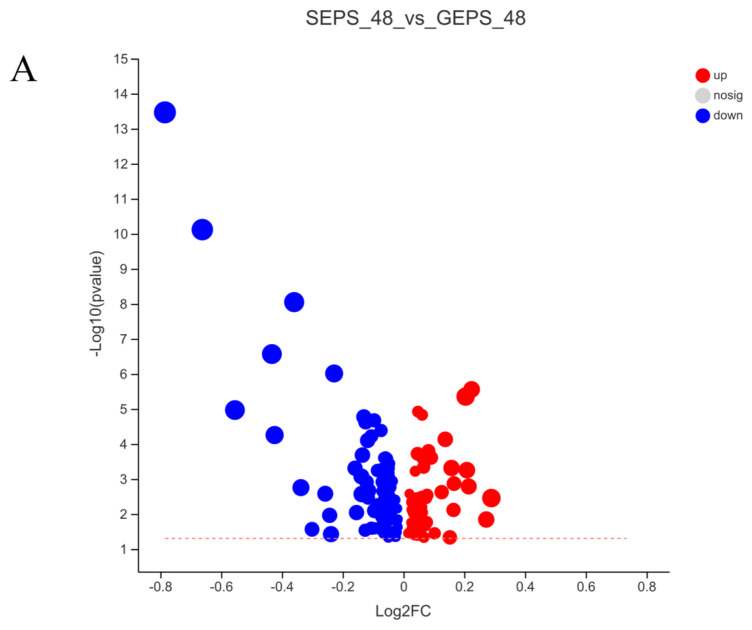
Analysis of DEMs in SEPS_48 and GEPS_48 comparison groups. (**A**) Number of up− and downregulated DEMs in the SEPS_48 and GEPS_48 comparison groups. Red dots represent upregulated metabolites, blue dots represent downregulated metabolites. (**B**) Heatmap of the DEMs in the SEPS_48 and GEPS_48 comparison groups. Red indicates highly expressed genes and blue indicates less−expressed genes.

**Figure 5 foods-11-03706-f005:**
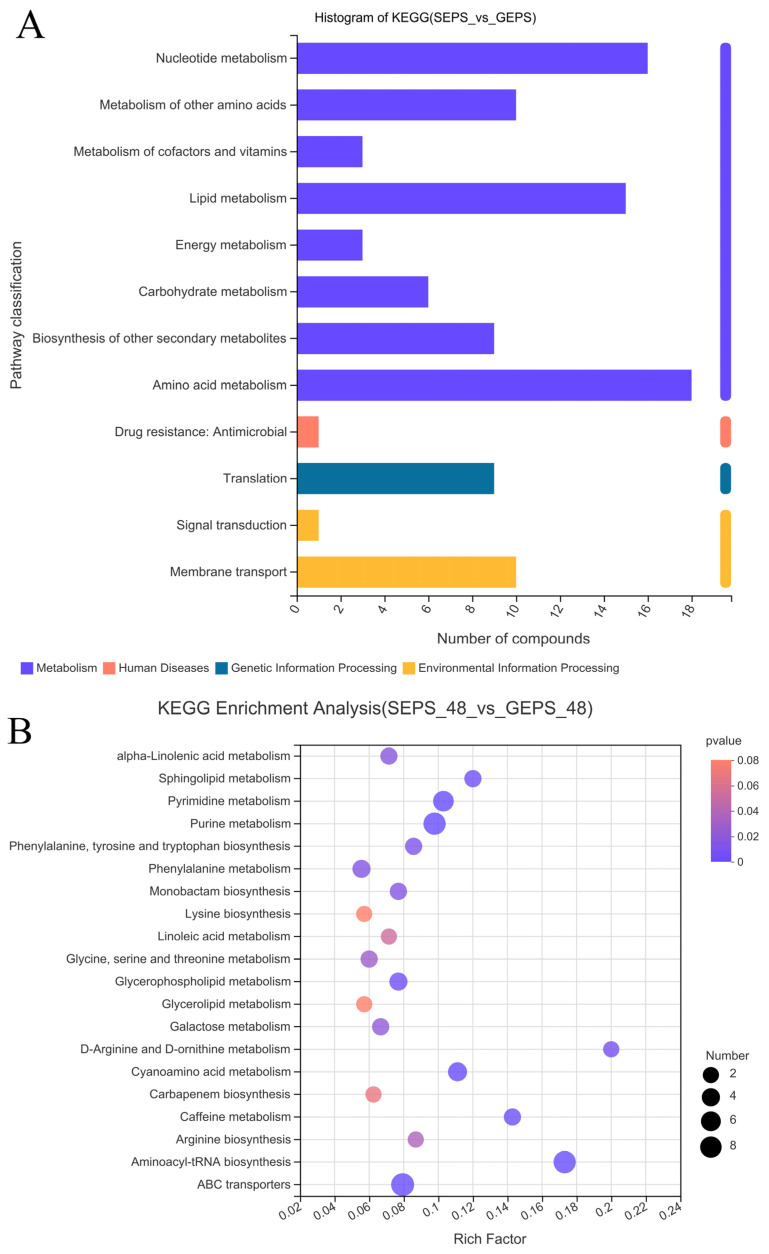
(**A**) KEGG annotation histogram of DEMs. (**B**) Enrichment analysis of DEMs between SEPS_48 and GEPS_48 comparison groups based on KEGG pathway. The color represents the *p*-value, and the size of the bubble represents the number of DEMs.

**Figure 6 foods-11-03706-f006:**
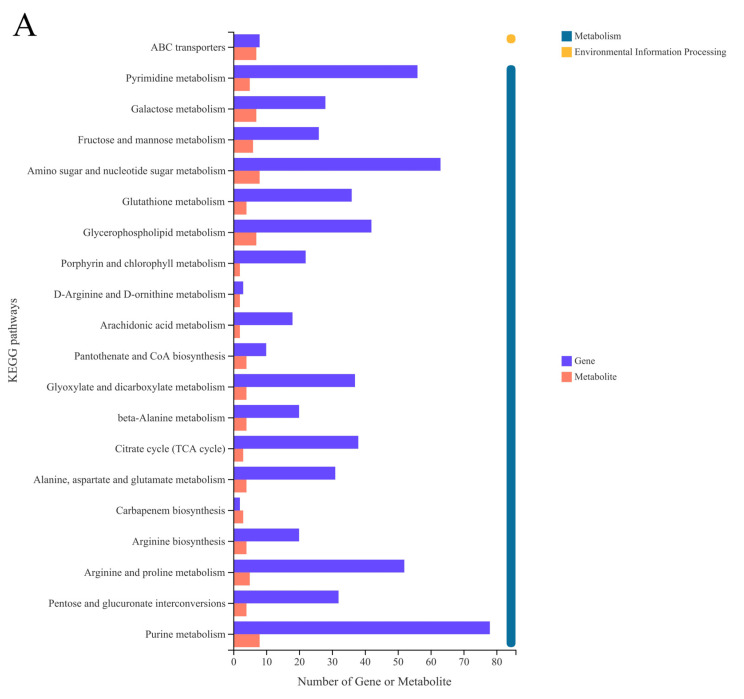
Correlation analysis of DEGs and DEMs between SEPS_48 and GEPS_48 comparison groups. (**A**) Correlation analysis of SEPS_48 and GEPS_48 KEGG pathway annotation histogram (the color represents pathways, and the size of the bubble represents the number of enriched genes or metabolites). (**B**) Heat map correlation analysis between DEGs and DEMs. *, *p* < 0.05. **, *p* < 0.01. ***, *p* < 0.001. The red and blue depths represent the strength of the positive and negative correlations, respectively. (**C**) Correlation network of DEGs and DEMs. Triangle represents metabolites, circle represents genes, connecting line represents correlation coefficient, red line represents positive correlation (correlation coefficient > 0), green line represents negative correlation (correlation coefficient < 0), and the thickness of the connecting line corresponds to the absolute value of the correlation coefficient.

**Figure 7 foods-11-03706-f007:**
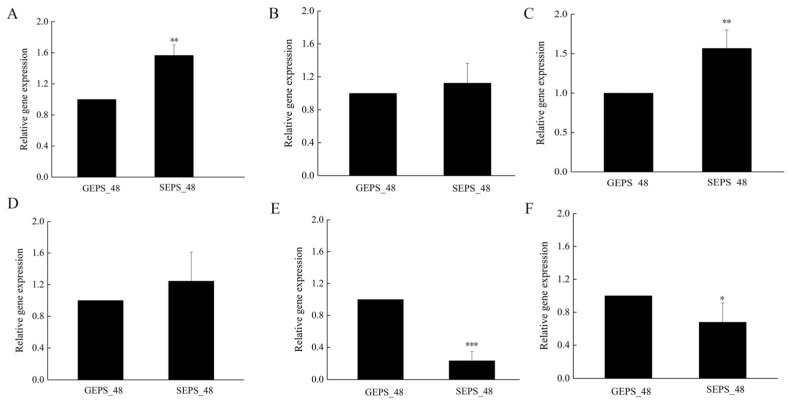
Analysis of relative gene expression level of some identified DEGs. (**A**) *pold3*. (**B**) *gpt*. (**C**) *cpsB*. (**D**) *uap*. (**E**) *ndk*. (**F**) *ure*. (These values are the mean ± standard deviation of three independent variables. *, *p* < 0.05. **, *p* < 0.01. ***, *p* < 0.001.)

## Data Availability

The data presented in this study are available upon request from the corresponding authors.

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
