# Peer review of "Transcriptomics and Metabolomics Analysis of Sclerotium rolfsii Fermented with Differential Carbon Sources"

_foods, 2022, doi:10.3390/foods11223706_

Round 1
Reviewer 1 Report
The manuscript is well written and is well presented. The work is necessary to understand the biosynthetic pathway of one and other fungal sources.
The author is suggested to refine the introduction to make it short crisp and to the point.
Also a thesis statement is must, the introduction seems to have one, the conclusion however does not state that it has been achieved, therefore align your manuscript around achieving a thesis statement.
Reviewer 2 Report
Revision of the manuscript foods-1992788
Note: please carefully check the English. I’m not a native speaking but it is evident that many sentences are poorly constructed and the meaning is not understood
The abstract must be revised: in the first sentence the verb is missing. The meaning of the acronyms DEGs, DEMs, SEPS_48 and GEPS_48 should be reported in parentheses.
introduction
Line 42: “which makes it can be used in several technologies, widely used in food processing …” change in “which makes it usable in several technologies, and then widely used in food processing…”;
Line 57: “muti-omics” change with “multi-omis”
lines 74-82: “When stud-74 ying the effects…” this paragraph should be rewritten the construction of the discourse is not clear
lines 86-88: “So as to provide…. by microorganism” also in this sentence the construction of the discourse is not clear.
Materials and Reagents
Lines 93-96 - check punctuation and sentence setting
Lines 117-118: indicate in brackets the meaning of SEPS_48 and GEPS_48; I guess that 48 is the growing time (48 h) but it’s never specified
Lines 118-119: “After the total RNA samples was qualified, the fungal samples needed to use magnetic beads with Oligo (dT) to purify the mRNA in Toltal RNA.” This sentence must be rewritten because it’s not clear and the English is not corret.
Lines 124-125: “To collect the target fragments and perform PCR amplification to complete the library construction. Finally, the tools such as fastx_toolkit to sequence the constructed library.” These sentences must be rewritten because they’re not clear and the English is not corret.
Line 126: “TGICL was used to…” change in “TGICL software was used to…”
Line 153: I think that the thermal cycle is incomplete or incorrect.. the elongation step is missing… please check
Results
Figure 1, 2, 4, 5 and 6: increase the size of the figures as well as the size fonts in it. It is impossible to read what is written in. Also improve the quality of the images. I suggest to put the three panel A, B and C in vertical sequence instead of horizontal line.
Figure 3 and 7: increase the size along the axes and in the legends.
Discussion
The discussion is well written and articulated and seems to report a correct interpretation of the results. However, the authors often refer to tables in the supplementary materials section, which are not included in the manuscript, so I could not verify that what was reported was correct
Reviewer 3 Report
Dear Editors,
The article entitled “Transcriptomics and metabolomics analysis of Sclerotium rolfsii fermented with differential carbon sources” is interesting and novel in which sucrose, glucose and lactose were used for fermenting Sclerotium rolfsii. In terms of the quality of the paper, I would recommend minor revisions, particularly for the following points:
1. Please explain what all the acronym in the abstract stands for.
2. Please convert all of the unit of rpm to g
However, I am not sure that this article is suitable for this journal as this research seems out of the scope of the journal. Alternatively, the authors must emphasize and elaborate the application of the fermented Sclerotium rolfsii for food application,
Reviewer 4 Report
- L37 check the Umb... and Poly... should it be cabs?
- Introduction lacks the purpose of the study; please clearly specify it.
- What is the outcome after understanding the transcriptomics and metabolomic of Scelerotium rolfsii?
- Sucrose, glucose, and lactose is it enough as a different carbon source? what about fructose, and amylose etc?
- Why did the author not tested on the crude food (can be fruit juice) as a reference to check the performance against the individual carbon source?
- Statistical section should be included in the text.
- All figures are too small, and difficult to see the results and cross-check the results with the text.
Round 2
Reviewer 2 Report
2nd Revision of the manuscript foods-1992788
General considerations
The authors followed the suggestions and made the requested changes in all sections of the paper.
The size of the and quality of images have been improved but they can be even more so.
Other correction required
Line 48 (line 42 in the 1st version): “muti-omics” change with “multi-omis”
Line 50-53: “It has been reported that the homopolysaccharides such as glucan [12, 13], fructan [14], and the heteropolysaccharides such as hyaluronic acid [15, 16] and xanthan gum [17], most of them are bacterial polysaccharides.” Change with:
“Several bacterial polysaccharides have been reported and characterized, such as glucan [12, 13] and fructan [14] homopolysaccharides, and hyaluronic acid [15, 16] and xanthan gum [17] heteropolysaccharides.”
Line 141: “The thermal cycle curve is as follow: 95 °C, 30s, 95 °C, 10s, 60 °C, 30s, 95 °C, 15s, 60 °C, 1min, 95 °C, 15s”
change with: “The thermal cycle curve was: step 1, denaturation at 95 °C/30s; step 2, PCR (40 repeats) 95 °C/10s, 60 °C/30s; step 3, melting at 95 °C/15s, 60 °C/1min, 95 °C/15s”
Line 149: “data to effectively reduced the influence” change with “data effectively reduced the influence”
Reviewer 4 Report
The authors have revised the paper according to the review comments, this paper can be accepted for publication.
